# *CHAMP1*-Related Disorder: Sharing 20 Years of thorough Clinical Follow-Up and Review of the Literature

**DOI:** 10.3390/genes14081546

**Published:** 2023-07-28

**Authors:** Sarah Abi Raad, Vanda Yazbeck Karam, Eliane Chouery, Cybel Mehawej, Andre Megarbane

**Affiliations:** 1Department of Pediatrics, UPMC Children’s Hospital of Pittsburgh, Pittsburgh, PA 15224, USA; sarah.abiraad@chp.edu; 2Department of Anesthesiology, Gilbert and Rose-Marie Chagoury School of Medicine, Lebanese American University, Byblos 1102-2801, Lebanon; vanda.abiraad@lau.edu.lb; 3Department of Human Genetics, Gilbert and Rose-Marie Chagoury School of Medicine, Lebanese American University, Byblos 1102-2801, Lebanon; eliane.choueiry01@lau.edu.lb (E.C.); cybel.mehawej@lau.edu.lb (C.M.); 4Institut Jérôme Lejeune, 75015 Paris, France

**Keywords:** intellectual disability, MRD40, *CHAMP1*, metabolic syndrome, neurodevelopmental delay

## Abstract

Intellectual disability (ID) is a prevalent neurodevelopmental disorder characterized by limitations in intellectual functioning and adaptive behavior. While the causes of ID are still largely unknown, it is believed to result from a combination of environmental exposures and genetic abnormalities. Recent advancements in genomic studies and clinical genetic testing have identified numerous genes associated with neurodevelopmental disorders (NDDs), including ID. One such gene is *CHAMP1*, which plays a role in chromosome alignment and has been linked to a specific type of NDD called CHAMP1 disease. This report presents the case of a 21-year-old Lebanese female patient with a de novo mutation in *CHAMP1*. In addition to ID and NDD, the patient exhibited various clinical features such as impaired language, dysmorphic features, macrocephaly, thoracic hyperkyphosis, decreased pain sensation, and metabolic syndrome. These findings expand the understanding of the clinical spectrum associated with *CHAMP1* mutations and highlight the importance of comprehensive follow-up for improved prognosis. Overall, this case contributes to the knowledge of CHAMP1-related NDDs by describing additional clinical features associated with a *CHAMP1* mutation. The findings underscore the need for accurate diagnosis, thorough follow-up, and personalized care for individuals with *CHAMP1* mutations to optimize their prognosis.

## 1. Introduction

Intellectual disability (ID) is a neurodevelopmental disorder (NDD) characterized by limitations in intellectual functioning and adaptive behavior [1]. It affects approximately 2 to 3% of the general population [2]. It typically emerges during childhood, but its clinical presentation is heterogeneous, often coexisting with congenital malformations or other NDDs such as epilepsy and autism spectrum disorder (ASD) [3]. While the causes of ID are not fully understood, they can be attributed to environmental exposures and/or genetic abnormalities [4]. Maternal exposure to a toxin, infectious agent, uncontrolled maternal condition, and birth complications represent the main identified environmental factors involved in ID. Regarding the genetic component, advancements in large-scale genomic studies and clinical genetic testing have resulted in the identification of a high number of specific genes that contribute to NDDs [5]. A positive molecular diagnosis is now reachable in more than 50% of individuals with ID and/or NDD, and approximately 30% in those with ASD, using high throughput sequencing approaches combined with chromosomal microarray analysis [6,7,8,9,10]. Till now, more than 2643 genes or loci have been linked to NDDs (https://panelapp.genomicsengland.co.uk/panels/, 21 June 2023). Several biological pathways involved in the physiopathology of these diseases have also been elucidated; including processes involved in synaptogenesis, chromatin remodeling, transcriptional regulation and chromosome alignment and spindle assembly [11,12,13,14]. Lately, a new gene involved also in proper chromosome segregation has been linked to a new type of NDD called CHAMP1 disease (or Mental Retardation 40, MRD40, MIM# 616579) [15]. This gene was named CHAMP1 for “Chromosome Alignment Maintaining Phosphoprotein 1”, also called “CAMP, ZNF828, or C13orf8”. It was detected to be mutated in two patients with severe developmental delay in the Deciphering Developmental Disorders (DDD) study [16], and later on, in 27 cases identified by GeneDX and at Ambry Genetics [14,17]. This disorder that was previously known as NS-ID (Orphanet no. 178469) [15] is now considered as a rare NDD, potentially accounting for only 0.03% of the cases [14]. As more cases with MRD40 have been reported in the past few years, a better understanding of the disease is expected in the future. Genetic testing remains the only method to diagnose MRD40 due to the overlapping of the neurological anomalies and dysmorphic features with other NDDs

Although expensive in some countries, genetic early diagnosis represents a key aspect of clinical management. Through the identification of genetic abnormalities and predispositions at the earliest stages of life, it helps healthcare professionals to intervene and implement targeted treatments. The importance of genetic diagnosis lies thus in its ability to detect hereditary diseases before symptoms manifest, enabling timely and precise interventions. The generated information from a genetic testing empowers thus patients to take charge of their health and make proactive choices to minimize risk factors in their families.

Here, we report a 21-year-old Lebanese female patient with a de novo mutation in *CHAMP1*. In addition to NDD, ID, impaired language, and dysmorphic features, the patient presented with macrocephaly, thoracic hyperkyphosis, and metabolic syndrome. WES performed in the patient enabled the identification of a de novo heterozygous nonsense known pathogenic variant p.Arg398* in *CHAMP1*. Our findings enable broadening the clinical spectrum linked to *CHAMP1* mutations by identifying new features, and highlights the importance of thorough follow-up for improving the prognosis of affected patients.

## 2. Material and Methods

### 2.1. Patient

We herein describe a 21-year-old woman with developmental delay and dysmorphic features born to a non-consanguineous Lebanese family, referred to our clinic for clinical and genetic evaluation.

### 2.2. Isolation of Genomic DNA

Written informed consent was obtained from legally authorized representatives of the patients (parental consent) to participate in this study and its publication. EDTA blood sample from the patient was collected for genetic studies. DNA was extracted from leucocytes by standard salt-precipitation methods [18].

### 2.3. Whole-Exome Sequencing (WES)

WES was carried out in the patient as mentioned previously [19]. Briefly, exome was captured and enriched using the solution Agilent SureSelect Human All Exon kit version 5.0. Concentrations of the prepared libraries were measured with Qubit Flex (Life Technologies, Carlsbad, CA, USA) using the dsDNA HS Assay Kit. The quality of the prepared libraries was assessed using Bioanalyzer 2100 with the High Sensitivity DNA kit (Agilent Technologies, Santa Clara, CA, USA). Samples were then multiplexed and subjected to sequencing with the average coverage of 100× on an Illumina HiSeq 2500 PE100-125. Reads files (FASTQ) were generated from the sequencing platform via the manufacturer’s proprietary software. Reads were aligned to the hg19/b37 reference genome using the Burrows–Wheeler Aligner (BWA) package version 0.7.11 [20]. SAM files were converted into BAM files and sorted using SAMtools v1.9 to check the percentage of the aligned reads (http://samtools.sourceforge.net, accessed on 29 May 2020). Variant calling was subsequently performed using the Genome Analysis Tool Kit (GATK) version 3.3 [21]. Variants were called using high stringency settings and annotated with VarAFT software 1.61 [22] containing information from dbSNP147 and the Genome Aggregation database (gnomAD, http://gnomad.broadinstitute.org, accessed on 8 June 2020). Only nonsynonymous coding and splicing variants found in the patient were considered. Variant filtering was performed according to the mode of transmission of the disease in the family, the frequency of the variant in the gnomAD database (<0.01% and <50 heterozygous carriers or <5 homo-/hemizygous carriers), and in our in-house database (<1 homozygous occurrence).

### 2.4. Sanger Sequencing

The genomic sequence of *CHAMP1* ((NM_001164144.1) was obtained from UCSC Genomic Browser. Primers used for PCR amplification were designed using Primer3 software (http://frodo.wi.mit.edu, accessed on 15 June 2020) to amplify the exon 3 of the *CHAMP1* gene including the p.Arg398* (c.1192C > T) detected by WES in the patient. PCR reactions were performed using Taq DNA polymerase (Invitrogen Life Technologies, Carlsbad, CA, USA). PCR fragments were run on 1% agarose gel. The fragments were purified using «SIGMA-ALDRICH TM» kit and then sequenced using the Big Dye_Terminator v1.1 Cycle Sequencing Kit (Applied Biosystems, Foster City, CA, USA). Sequence reaction was purified on Sephadex G50 (Amersham Pharmacia Biotech, Foster City, CA, USA), and then loaded into an ABI3500 system after the addition of Hidi formamide. Electropherograms were analyzed using Sequence Analysis Software version 5.2 (Applied Biosystems) and then aligned with the reference sequences using ChromasPro v1.7.6.1 (Technelysium, South Brisbane, QLD, Australia).

## 3. Results

### 3.1. Clinical Description

The patient described herein is a 21-year-old female patient born as a fourth child to non-consanguineous parents of Lebanese descent. The family history was unremarkable for any neurodevelopmental disorder. The patient’s mother was 37 years old at the time of conception and the father 50 years old. The pregnancy proceeded without any complications and the delivery was conducted via elective repeat cesarian section at 38 weeks and 3 days, resulting in favorable APGAR scores of 8 at 1 min and 10 at 5 min. The patient’s birth measurements indicated a weight of 3.3 kg (+0.5 SD), height of 50 cm (+0.5 SD), and an occipital frontal circumference (OFC) of 33 cm (−0.7 SD) (per WHO growth charts).

Distinctive facial characteristics (Figure 1), including a round face, a flat head with a shortened forehead, hypertelorism, upslanted palpebral fissures, and strabismus were identified in the patient. At the age of 4 months, parents noticed generalized hypotonia, drooling, and a blank stare, prompting a comprehensive evaluation. A skeleton survey was performed and found to be normal. Laboratory results revealed mildly elevated levels of lactic acid and pyruvate, while the urine organic amino acid panel and thyroid-stimulating hormone (TSH) levels remained within the normal range. An MRI performed at 5 months of age indicated small vermian hypoplasia and peri-cerebral effusion, with no abnormalities detected in the gray and white matters. An EEG, conducted at the age of 6 months to investigate abnormal movements during sleep onset, yielded normal results. She started early intervention therapies including speech therapy, occupational therapy, psychomotor therapy, psychotherapy, and one-to-one special education sessions to improve her social integration. Although the patient achieved developmental milestones such as rolling over, crawling, and walking within expected time frames, she had difficulties adapting her muscle tone with respect to required fine and gross motor tasks. For instance, her walk was characterized by a stepping gait and stiff body. She exhibited a short attention span and impaired visual pursuit, which further compromised her poor fine motor skills. She was late to use a pencil, scissors, and feed self. Milestones were also reportedly late in areas of language. Delays predominantly impacted expressive language, with notable challenges in articulation. She spoke her first words when she was approximately 1 year and 6 months old and put a sentence together at approximately age 3. While receptive language was relatively less affected, difficulties in speech and communication were evident. Occasional temper tantrums were observed, likely arising from a gap between comprehension and expression, as well as struggles verbalizing emotions. She also exhibited stereotypical behaviors, such as finger crisping accompanied by facial muscle tension. Overall, the patient consistently displayed endearing and friendly behavior, often with high energy levels.

A brain MRI at age 4 revealed focal prominence of sulci over the posterior aspect of the right parietal lobe, suggesting the possibility of post-traumatic encephalomalacia, atrophy, or a small arteriovenous malformation (AVM). At age 6, an EEG revealed mild diffuse encephalopathy without epileptiform activity.

During the next few years, the patient was seen sporadically for routine clinical check-ups. She was in good health, with no major complaints. Despite having an ID, she has attended a regular school with the aid of a shadow teacher and a customized curriculum. Owing to challenges with fine motor skills and limited dexterity for handwriting, the patient has successfully adapted to using a keyboard as her primary means of written communication. Speech was intelligible to most. Encouragingly, she has demonstrated greater progress than anticipated in activities of daily living, though continued efforts are necessary to enhance her level of independence. Thoracic hyperkyphosis was diagnosed at the age of 13 years that necessitated wearing of a spine brace and regular physical therapy.

A formal Neuropsychological Assessment was performed at the age of 14 years. General intellectual functioning (i.e., IQ = 40) was below typical age expectations. Within that context, weaknesses across cognitive domains were found. Verbal memory was found to be an area of strength in her profile. Cognitive weaknesses impeded adaptive skills across several areas of functioning. Academically, she had basic reading and calculation skills. Diagnostically, based on test scores, adaptive functioning, behavioral observation, and history, the patient’s neuropsychological profile was consistent with a DSM-V diagnosis of moderate intellectual disability (ID).

As the patient transitioned into adulthood, additional diagnoses were made. Decreased pain sensation was noted. Overweight body habitus at age 15 with abdominal obesity were observed. Additionally, hypertriglyceridemia, low HDL cholesterol levels and impaired fasting glucose were found. Hypertension was discovered at the age of 16 necessitating the use of an angiotensin receptor blocker. All these findings suggest the diagnosis of metabolic syndrome. In addition, elevated uric acid levels and polycystic ovary syndrome (PCOS) were discovered at the age of 18 and necessitated the use of metformin. Moreover, a second-generation antipsychotic medication, Aripiprazol, was started at the age of 18 to address temper outbursts and mood swings.

Now, at the age of 21, her OFC is 58.5 cm (>+2 SD), her height 173 cm (+0.67 SD), and her weight 84 kg (>+2 SD), and a BMI of 28. She presents with macrocephaly, a wide nasal bridge, hypertelorism, narrow palpebral fissures, a bulbous nose, a small midface, full cheeks, a small philtrum, everted lips, a pointed chin, a short neck, and fifth finger clinodactyly. She experienced menarche at the age of 13. She remains well followed by a multidisciplinary team of therapists. She is an affectionate and friendly young woman. No ADHD (Attention-Deficit/Hyperactivity Disorder), nor autistic behaviors were noted. She shows clear social strengths, including maintaining appropriate eye contact, initiating interactions, engaging in social games, and displaying a diverse range of facial expressions.

### 3.2. Genetic Studies

WES was performed in the patient and led to the identification of approximately 99,459 variants. Filtering was then performed to exclude all non-genic, non-splice site, and intronic variants as well as all frequent variants (present in more than 1% in databases). This filtering strategy led to the selection of a list of 651 variants that were thoroughly studied in order to select the candidate variant that can explain the clinical picture of the patient. One known heterozygous variant, the p.Arg398* (c.1192C > T) in *CHAMP1* (NM_001164144.1), was considered as the only candidate variant.

The selected variant, located in the exon 3 of *CHAMP1*, is predicted to be pathogenic according to ClinVar and is classified as Class IV: PVS1, PP5 and PM2 based on the ACMG classification [23]. Further analysis by Sanger sequencing of the patient and his parents revealed a de novo occurrence of the variant.

Since a metabolic syndrome was diagnosed in the patient, WES data were reanalyzed but did not reveal any pathogenic variant that may explain the clinical signs related to the mentioned syndrome.

This adopted diagnosis strategy follows the workflow for diagnosing Mendelian diseases as per the DDD project recommendations [15]. The workflow includes three main steps: (i) obtaining sequencing data from the patient—including the parents (WES trio analysis) is preferred in order to determine the inheritance of the candidate genetic variants; (ii) assessment of the pathogenicity of the identified variant; and (iii) matching the phenotype.

## 4. Discussion

Here, we report the case of a 21-year-old woman, born to a non-consanguineous Lebanese family and presenting with ID, DD, impaired language, dysmorphic facial features, macrocephaly, thoracic hyperkyphosis, decreased pain sensation and metabolic syndrome. WES, a powerful tool for the diagnosis of NDD, performed in the patient enabled the identification of a heterozygous nonsense known pathogenic variant p.Arg398* in *CHAMP1*. The identified gene, located on chromosome 13q34, is known to be involved in a form of NDD, named NEDHILD, since it includes the following features: NDD with hypotonia, impaired language, and dysmorphic features. Hypotonia, microcephaly, seizures, ophthalmologic manifestations, constipation/gastroesophageal reflux, and behavioral issues, including autism and sleep disturbances are seen as well [17].

The pathomechanisms underlying CHAMP1 disorder are still poorly understood. CHAMP1 is a zinc finger protein consisting of 812 amino acids. It is expressed during fetal brain development as well as in all adult tissues [24]. The protein contains five C-terminal C2H2 zinc finger domains that are involved in binding to chromosomes on the mitotic spindle, thereby facilitating appropriate chromosome alignment. Kinetochore-microtubule attachment and chromosome segregation modulation, are both known to be important for neurodevelopment in Human [15].

*CHAMP1*-linked phenotype was described for the first time by Hempel et al., in 2015 in five patients presenting with ID, hypotonia, speech delay, dysmorphic facial features, and amicable and/or stereotyped behavior, in addition to decreased pain sensation, and mi-crocephaly in some of them [14,16] (Appendix A). In the next few years, patients with additional manifestations were reported; some with hearing loss or sleep disturbances [25,26] or gastrointestinal abnormalities [17]. Subsequently, there have been reports of cases involving obesity [16] and one case report of hyperinsulinemia in patients with *CHAMP1* mutations [27], thus suggesting the possible presence of an underlying metabolic syndrome in *CHAMP1* patients that was not highlighted till date. Indeed, metabolic syndrome, as defined by the National Heart, Lung, and Blood Institute (NHLBI), encompasses a cluster of metabolic factors, including abdominal obesity, high blood pressure, impaired fasting blood glucose, high triglyceride levels, and low HDL cholesterol [28]. A diagnosis of metabolic syndrome is typically made when a person presents with three or more of these factors. In this article, we report the first documented case of a patient with CHAMP1 disorder exhibiting all the features of metabolic syndrome.

Of note, the PCOS as well as the metabolic syndrome identified in this patient might be a fortuitous association especially with the high prevalence of PCOS in the MENA (Middle East and North Africa region) region [29]. Metabolic syndrome is also highly prevalent in this region [30], but in older individuals. Our patient was diagnosed at the age of 16 with hypertension, which is considered an early age compared to the studied population with metabolic syndrome. Nevertheless, investigating the presence of any metabolic syndrome in cases with *CHAMP1* mutations is needed for a better follow-up and clinical management of this population.

Importantly, while most of the cases described until now [14,16,17,24,25,26,27,31,32] (Appendix A) present either with microcephaly (22/45) or normal head circumference (23/45), the patient described in this paper as well as only another patient identified by Levy et al. in 2022 [16] present with macrocephaly, illustrating once more the clinical heterogeneity of this disease and highlighting the challenges encountered in the clinical diagnosis of patients affected with NDD. Furthermore, 10 out of the 45 cases described in the literature (Appendix A), in addition to our patient showed high sensitivity to pain. This essential finding should be seriously considered by genetic counselors and/or pediatricians. Indeed, sharing this information with the parents or legal guardians is crucial in order to avoid any self-severe injury in the patients with *CHAMP1* mutations.

Last but not least, two patients described by Hempel et al., in 2015, the first Dutch (C:II-2) and the second German (E:II-2) share the same nonsense mutation, p.Arg398* with the patient described herein. Patient C:II-2 presents with severe ID while E:II-2 was described as having moderate ID. Our patient was followed-up thoroughly by her family and with the help of a multidisciplinary team of therapists since early age. Despite having ID, she was able to attend a regular school with the aid of a shadow teacher and a customized curriculum. She has also successfully adapted to using a keyboard as her primary means of written communication. Her speech is now comprehensible to most and she is demonstrating greater progress than anticipated in the activities of daily living. This extraordinary progress highlights the importance of a thorough follow-up since early detection of clinical signs in order to offer these patients a better quality of life.

Indeed, early childhood is a pivotal period for optimal brain growth and development, yet it is also a vulnerable phase. The timing of factors impacting brain development is of utmost importance, as there are specific early windows of opportunities. Failing to harness these may hinder optimal brain development and lifelong well-being [33]. Therefore, to support the development of children diagnosed with disabilities, early childhood interventions are essentials. These involve a range of coordinated services, such as clinic-based care, school-based programs, parenting support, and home-based therapies. They have been proven to prevent significant declines in intellectual development that often occur during early childhood for children with developmental delays. It also positively impacts the child’s long-term developmental trajectory, reducing the risk of secondary health and psychosocial complications. Implementing family-centered interventions has also shown to enhance the psychosocial well-being of both the child and caregiver [34]. However, timely identification of children with developmental disabilities is crucial for the success of early childhood interventions programs.

Of note, two main approaches are essentials for early childhood interventions for children with disabilities. Firstly, integrating them into mainstream services for Early Childhood Development interventions and secondly implementing targeted intervention programs. These approaches vary significantly to cater to the diverse needs of children and families.

Importantly, although comprehensive early intervention services for children with disabilities are prioritized in high-resource countries such as the United States, the situation is different in lower- and middle-resource countries [35]. Inadequate country health systems present a significant obstacle to providing high-quality services. Additionally, cultural challenges, including stigma and discrimination surrounding children with disabilities and their families, pose further difficulties. Although it is essential to include children with developmental disabilities in mainstream services or specialized early childhood interventions or both, families of these patients play a critical role in filling the existing gaps in service availability in these countries. 

Our patient underwent an early comprehensive assessment and diagnosis, which allowed to customize intervention and support to address her specific needs. She had the chance to receive a hybrid approach that combines inclusive education since the age of 1 year, aided by a shadow teacher. Additionally, she has benefitted from a focused and targeted intervention program that meets her special needs, starting from the early diagnosis at 5 months and continuing through her current age, with the help of various intervention therapists. As mentioned before, the thorough follow-up that was offered for the patient described herein manifested in an extraordinary progress of her developmental milestones.

On another note, WES analysis in this patient showed once more its successful application in discovering of the gene associated with a Mendelian phenotype. Additionally, this latter finding stresses the value of genetic diagnosis and counseling. Indeed, with the identification of the genetic aberration involved in the disease of the patient reported herein, genetic counseling for the siblings as well as for the entire family has become possible. Genetic counseling serves as a crucial support system for individuals with a family history of genetic disorders. Through genetic counseling, professionals provide comprehensive information about genetic conditions, the implications of test results, and available options for managing or preventing genetic risks. By promoting informed decision-making and emotional support, genetic counseling not only enhances the quality of life but also contributes to the prevention and early detection of genetic conditions, leading to better healthcare outcomes for individuals and their families.

In conclusion, our findings enable broadening the clinical spectrum linked to *CHAMP1* mutations by adding new features and highlight the importance of an early diagnosis and thorough follow-up for a better prognosis of affected patients.

## Figures and Tables

**Figure 1 genes-14-01546-f001:**
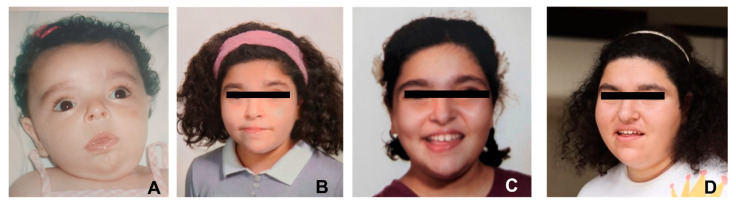
Pictures of the patient at different ages. From left to right: Age (**A**) 4 months; (**B**) 5 years; (**C**) 12 years; and (**D**) 21 years.

## Data Availability

The datasets used and analyzed during the current study are available from the corresponding author upon a reasonable request.

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
