# Peer review of "CHAMP1-Related Disorder: Sharing 20 Years of thorough Clinical Follow-Up and Review of the Literature"

_genes, 2023, doi:10.3390/genes14081546_

Round 1

Reviewer 1 Report

This article describes a single case of intellectual disability associated with the CHAMP1 gene mutation. It contains all the necessary information about the etiopathogenesis and symptoms of this syndrome. This is a rarely described disease, so the advantage of the presented article is a 20-year period of observation of psychomotor development and comorbidities.

I do not understand why the authors cite their own work on autism associated with the mutation of the PCDH19 gene in this article.

There are references in the text to Table 1, which is not in the article.

----------

1. What is the main question addressed by the research? -The article describes a single case of intellectual disability associated with the CHAMP1 gene mutation. It contains all the necessary information about the etiopathogenesis and symptoms of this syndrome.

2. Do you consider the topic original or relevant in the field? Does it address a specific gap in the field?  -This is a rarely described disease, so the advantage of the presented article is a 20-year period of observation of psychomotor development and comorbidities.

3. What does it add to the subject area compared with other published material? -This is a typical case report with a comparison of differences in the phenotypes of patients described in other publications

4. What specific improvements should the authors consider regarding the methodology? What further controls should be considered? - The methodology is consistent and does not require corrections

5. Are the conclusions consistent with the evidence and arguments presented and do they address the main question posed? -The article includes descriptions of previously published cases and the conclusions are correctly presented. Differences in the course of the disease and dysmorphic features found in individual case reports are also discussed.

6. Are the references appropriate? -The authors cite their previous publication, which is not related to the issues discussed in this article

7. Please include any additional comments on the tables and figures. There are references in the text to Table 1, which is not in the article.

Author Response

Reviewer #1:

Q1: I do not understand why the authors cite their own work on autism associated with the mutation of the PCDH19 gene in this article.

R1: We would like to thank the reviewer for his input. The paper was cited in the "methods" section as it describes in more details the same WES methodology used in the current article.

Q2: There are references in the text to Table 1, which is not in the article.

R2: We would like to thank the reviewer for his comment. Both references were added to the text as per his suggestion (page 5 last paragraph).

Reviewer 2 Report

Excellent report. It deals with an important aspect of neurodevelopmental disorders, which is the phenotypic evolution with time.

Clinical data, including behaviour and cognitive abilities, are described in detail. 

Two minor criticisms
1) among unusual clinical manifestations, it is recommended to highlight macrocephaly and the metabolic disease; in the present form, decreased pain sensitivity is apparently decribed as a new finding

2) please define measurements in SD rather then > 97th centile

Author Response

Reviewer #2:

Two minor criticisms
Q1) among unusual clinical manifestations, it is recommended to highlight macrocephaly and the metabolic disease; in the present form, decreased pain sensitivity is apparently described as a new finding

R1: We would like to thank the reviewer for his valuable comment. This was checked and corrected in the entire text accordingly.

Q2) please define measurements in SD rather than > 97th centile

R2: We would like to thank the reviewer for his valuable comment. This was adjusted in the entire text.